# Direct and Indirect Influences of Objective Socioeconomic Position on Adolescent Health: The Mediating Roles of Subjective Socioeconomic Status and Lifestyles

**DOI:** 10.3390/ijerph16091637

**Published:** 2019-05-10

**Authors:** Concepción Moreno-Maldonado, Pilar Ramos, Carmen Moreno, Francisco Rivera

**Affiliations:** 1Department of Developmental and Educational Psychology, University of Seville, 41018 Seville, Spain; pilarramos@us.es (P.R.); mcmoreno@us.es (C.M.); 2Department of Experimental Psychology, University of Seville, 41018 Seville, Spain; franciscorivera@us.es

**Keywords:** adolescence, health, healthy lifestyle, objective socioeconomic position, subjective socioeconomic status

## Abstract

The use of composite indices and subjective measures to evaluate socioeconomic position, taking into account the effect of inequalities on adolescent health-related behaviors, can contribute to understanding the effect of inequalities on health during adolescence. The aim of this study was to examine the direct and indirect contribution of objective and subjective socioeconomic factors in a broad range of health and lifestyles outcomes. The data come from a representative sample of adolescents (*N* = 15,340; M age = 13.69) of the *Health Behavior in School-aged Children* study in Spain. Structural equation modeling was used for data analysis. A global index for evaluating objective socioeconomic position predicted both health and healthy lifestyles. Subjective socioeconomic status mediated the relationship between objective socioeconomic position and health but did not have a significant effect on healthy lifestyles when objective indicators were considered. Lastly, fit indices of the multiple-mediator model—including the direct effect of objective socioeconomic position on health and its indirect effects through the subjective perception of wealth and lifestyles—explained 28.7% of global health variance. Interventions aimed at reducing the impact of health inequalities should address, in addition to material deprivation, the psychological and behavioral consequences of feeling poor.

## 1. Introduction

The model proposed by the Commission on Social Determinants on Health (CSDH) [1] represents one of the most appropriate conceptual frameworks for studying social determinants on health and the relationships that are established between them. Following this model, there are two groups of social determinants of health: structural and intermediary. The structural determinants include social, economic, political, and cultural factors that determine education, occupation, and income, and therefore generate social stratification according to socioeconomic position. The intermediary social determinants of health are material conditions (such as living conditions, food availability, neighborhood), behavioral (nutrition, physical activity, or smoking), and psychosocial circumstances (psychosocial stressors, social support, copping strategies). Therefore, following the CSDH model, the socioeconomic and political context define a social hierarchy that includes factors such as social class, power, prestige, or discrimination that are unequally distributed across socioeconomic position, gender, or ethnicity. Thus, socioeconomic position crystalizes structural determinants of health, and influences an individual’s health and wellbeing through the intermediary factors.

In recent decades, scientific research has demonstrated the influence of socioeconomic determinants on health [2]. However, despite its importance, socioeconomic inequalities in adolescent health have been discussed infrequently and less in-depth [3], and the limited research often shows contradictory results. The lack of association between socioeconomic position (SEP) and adolescent health has been explained by the *equalization hypothesis,* which states that health inequalities tend to diminish during this period [4,5]. Nevertheless, evidence of SEP inequalities in health outcomes such as self-perceived health, frequency of psychosomatic symptoms, or mental health have also been found in adolescents [6,7,8]. 

There are various explanations for these contradictory results, and also for explaining the absence of a relationship between socioeconomic position and health. Firstly, the measures employed to evaluate SEP are inconsistent [9], and a wide range of variables have traditionally been used, principally income, educational level, and occupational status [10]. However, parental income cannot be reported by the adolescents themselves. In addition, missing values and inaccurate information provided by the adolescents about their parents’ educational and occupational status have raised certain problems [11]. The Family Affluence Scale (FAS) (based on the family’s material possessions) is a validated measure for evaluating adolescent SEP, developed within the framework of the *Health Behavior in School-aged Children* (HBSC) study with the purpose of solving some of these problems [12]. However, researchers agree that each socioeconomic indicator evaluates a different dimension of the SEP construct and provides only partial information about the resources available to a person [10], and therefore the use of other SEP measures are recommended in conjunction with FAS [13]. In this regard, many researchers have opted to use composite indices to capture the multidimensionality of SEP [14,15,16]. In order to address this methodological issue, an Index of Objective Socioeconomic Position (IOSEP) has been proposed to evaluate objective adolescent SEP based on parental education and occupation, in addition to FAS, and has demonstrated a more accurate health prediction than the measures employed individually [17]. 

Secondly, socioeconomic position refers not only to all the aforementioned objective dimensions, but also represents a subjective dimension of inequality, having been demonstrated that health is affected not only by the scarcity of material resources but also by the perception of a low socioeconomic status [18]. Subjective socioeconomic status (SSS)—capturing the individual’s experiences as a member of a specific socioeconomic group—is an easy indicator to measure and presents low rates of missing values [19]. This experience is strongly related to positive self-esteem during adolescence [20]. In addition, previous findings have reported certain consequences on adolescent health of perceiving oneself as having a low SEP in comparison to others, even after controlling the effect of objective indicators [21,22,23]. Moreover, a study conducted by Ahlborg et al. [24] demonstrated that controlling either the objective or subjective factors might affect the others impact on health. For example, the effect of FAS on multiple health complaints diminished or even reverted when SSS was controlled. Nonetheless, the interaction between these two types of indicators has been less explored, and although previous research has found low correlations between objective and subjective SEP measures [25,26], their independent or combined relationship with health remains unclear. 

Thirdly, the specific health outcomes examined might explain the equalization and inconsistent findings regarding the impact of inequalities during adolescence. In fact, equalization is more present in some specific measures of health or healthy lifestyles whereas not in others [4,26]. Moreover, some researchers argue that the long-term health effects of early adverse experiences have yet to manifest themselves during adolescence, and therefore measuring lifestyles, rather than health directly, is a more adequate evaluation of how socioeconomic inequalities affect adolescents [5,27].

Returning to the CSDH model and focusing on the mechanism through which socioeconomic position influence health, three main paths were proposed. The first mechanism explains health inequalities as the result of the differential exposure to experiences that are determined by material conditions. Therefore, although material wealth does not have a direct effect on health, it facilitates access to material resources that promote it. The second mechanism is understood as the effect of health inequalities through psychological consequences of poverty, such as increasing stress and eroding social resources. Previous findings have demonstrated how inequalities exert their influence through psychological mechanism such as family, friends, and school factors during adolescence [7,28]. The subjective perception of wealth represents an indicator of subjective socioeconomic status but also captures the psychological perception of pertaining to a specific socioeconomic group. Lastly, the third mechanism explaining how socioeconomic position affects health proposed in the CSDH model focuses on its impact through health-related behaviors. As defined by the World Health Organization (WHO) [29], *health behaviors* constitutes “any activity undertaken by an individual, regardless of actual or perceived health status, for the purpose of promoting, protecting, or maintaining health, whether or not such behaviour is objectively effective towards that end”. *Risk behaviors* are conceptualized separately, as “behaviours associated with increased susceptibility to a specific cause of ill-health”. Finally, health behaviors and risk behaviors form complex patterns of behaviors that are defined by the WHO as *lifestyles*. In some ecological health models of the determinants of health, such as those proposed by Dahrlem and Whitehead [30], lifestyles are considered to be on an ecological level and to be the most proximal determinant of health. Lifestyle inequalities might therefore explain, at least partially, socioeconomic inequalities in adolescent health [7,27,31]. For example, Moor et al. [28] demonstrated that family affluence inequalities in self-perceived health were explained almost 50% by behavioral and psychological factors. However, similar to studies on health inequalities, an association has been found regarding health-related behaviors such as frequency of eating breakfast or tooth brushing [32,33,34], but findings are inconsistent regarding other behaviors such as substance abuse or use of leisure technology [35]. In addition, although socioeconomic inequalities may affect certain health behaviors more than others, especially during adolescence, findings also show a varying relationship between SEP and lifestyles depending on the objective or subjective socioeconomic indicator employed [33,36].

Therefore, a fourth explanation for the difficulty in establishing the relationship between SEP and health during adolescence could be the underestimation of its impact when examining only the directs effects and not considering all mechanisms through which SEP exerts its influence on health.

As discussed, results regarding the impact of SEP on adolescent health are difficult to compare due to differences in methodological approaches (such as employing different objective and subjective SEP indicators, health/lifestyle outcomes, statistical strategies) in addition to country and time differences. In order to better understand the relationship between socioeconomic position (SEP) and adolescent health, the source of socioeconomic health differences and the mechanisms that maintain them must be identified [37]. 

### Research Objectives

In this context, the present research has two aims: to explore the impact of objective and subjective socioeconomic indicators—independently and combined—on adolescent health (Models 1–3) and healthy lifestyles (Models 4–6); and to analyze the direct and indirect effects of objective socioeconomic position on adolescent health through the subjective perception of wealth and health behaviors (Model 7). All models explored are presented in Figure 1. It was hypothesized that: (1) Objective and subjective indicators are related to health and lifestyles and that—considering both types of indicators—higher inequalities will be estimated; (2) the subjective perception of wealth will mediate the association between objective SEP, and health and lifestyles; and (3) healthy lifestyles also mediate the relationship between SEP and health. 

## 2. Materials and Methods

### 2.1. Participants and Procedure

The data come from a representative sample of Spanish adolescents composed of 15,340 students aged 11–16 years old who participated in the 2014 edition of the HBSC study. Mean age of participant was 13.69 years (*SD* = 1.72) and the sample showed a balanced representation of boys and girls (50.7% girls). For further details about the data collection process, consult Moreno et al. [38].

Data collection abided by the following recommendations from the HBSC international coordination team [39]: The questionnaires must be self-reported and administered in the schools under teacher supervision, as well as guaranteeing the participant’s anonymity. 

### 2.2. Instruments

The instrument employed was the Spanish version of the 2014 HBSC questionnaire (available at https://www.mscbs.gob.es/profesionales/saludPublica/prevPromocion/promocion/saludJovenes/estudioHBSC/docs/Cuestionarios/HBSC2014_Cuestionario_Alumnado.pdf). The HBSC questionnaire is a standardized instrument composed of validated scales for measuring several aspects of adolescent health and wellbeing [39]. Key measures of socioeconomic aspects, health, and lifestyles were selected for this study. The specific variables and the instruments used to measure them are detailed in Table 1.

Two instruments were used to examine objective and subjective dimensions of adolescent SEP. To evaluate the adolescents’ objective SEP, a latent factor of five socioeconomic factors was drawn (IOSEP): educational level and occupational status of each parent and family affluence scale. This global index has demonstrated its unidimensional factorial structure and its validity for predicting adolescent health [17]. Alpha reliability in this study was 0.78, indicating good internal consistency. Subjective socioeconomic status was represented as manifest variable.

With respect to adolescent health, a global measure was employed. Health is a broad concept that, due to its complexity, is also difficult to measure. Many experts suggest that health should be assessed through integrated models that capture its different dimensions, including variables that evaluate not only physical health but also emotional wellbeing, and which include the subjective experience of health. Moreover, it has been recommended that health measures should not only include indicators of wellbeing, but also negative indicators such as psychopathological symptoms. Therefore, with the aim of capturing an integrated concept of health, a latent factor based on empirical evidence was also drawn (global health score) composed of four measures: life satisfaction, self-rated health, health-related quality of life, and psychosomatic complaints. This factorial score was chosen because it encompasses multiple aspects of physical and psychological health, has previously shown good psychometric properties, and has proven to be a reliable and valid measure of health [40]. The Cronbach’s alpha for the 20-item scale in this study was 0.79, indicating good internal consistency.

Lastly, considering the lack of unidimensionality among healthy lifestyles, no latent factor was drawn. However, also considering the advantages of employing global indices to capture multidimensional constructs, we created a scale based on a wide range of behaviors related to lifestyles that have been demonstrated to be key for adolescent health: frequency of breakfast, dietary habits (consumption of fruit, vegetables, sweets, and soft drinks), tooth brushing, physical activity (moderate-to-vigorous physical activity and vigorous physical activity), hours of sleep, tobacco use, and alcohol consumption. The scores of each variable were coded from the least healthy behavior (1) to the healthiest behavior (3), following scientific recommendations for each habit. Thus, the global score of healthy lifestyles varied from 11 (less-healthy lifestyles) to 33 (more-healthy lifestyles). The description of each variable and the criteria behavior classification as more or less healthy are detailed in Table 1. In this case, alpha reliability for the 11-item scale was 0.53 (this questionable value of the Cronbach’s alpha will be addressed in the Discussion). 

### 2.3. Data Analysis

Spearman’s correlation coefficient was used to analyze the association between variables. The correlation size was analyzed according to recommended criteria for behavioral sciences [57]: Values around 0.10 were considered low correlations, moderate correlations when the value reached 0.30, and highly correlated for values equal to or higher than 0.50. The statistics program IBM SPSS 22 was used for these analyses and to obtain the descriptive statistics of the variables. 

Several measurement models (Figure 1) were examined using structural equation models in order to analyze how socioeconomic inequalities exert their influence on health and lifestyles. Specifically, Confirmatory Factor Analysis (CFA) with latent factors (representing IOSEP and the global health score) were conducted using the statistics program Mplus version 7. The analyses were conducted using the robust maximum likelihood (MLR) estimation method given its applicability to non-normal data. Participants with missing data were included in the model estimation employing full information maximum likelihood (FIML) estimation [58] to avoid any bias in the analysis due to missing values. Chi-square (*χ*²) was employed to test the overall fit of the models, which should not be significant for adequate model fit [54]. In addition, the following indices were employed: Comparative Fit Index (CFI), Tucker–Lewis Index (TLI), Root Mean Square Error of Approximation (RMSEA), and Standardized Root Mean Squared Residual (SRMR). Values for the indices CFI and TLI above 0.90 suggest acceptable model fit and excellent fit at 0.95 or higher [59,60]. For the RMSEA, values are accepted close to or lower than 0.08 and 0.05 for SRMR [61]. 

The estimated models analyzed the independent effect of IOSEP on health (Model 1), the effect of the SSS on health (Model 2) and the combined effect of both types of socioeconomic indicators on health (Model 3). Secondly, the same models were performed employing healthy lifestyles as a dependent variable (Models 4, 5, and 6, respectively). Additionally, the Lagrange Multiplier (LM) test was performed, suggesting an improvement of Model 3 by adding a path between IOSEP and SSS (Model 3b), justified by empirical and theoretical evidence [62]. The re-specified model showed better fit with a significant decrease in Chi-square and an increase in CFI superior to 0.01 with respect to the previous Model 3 [63]. Lastly, a multiple mediator model was estimated, including the direct effect of IOSEP on health and the aggregated indirect effects of IOSEP on health through healthy lifestyles and SSS (Model 7). The total direct and indirect effects of the proposed models were estimated employing the bootstrapping procedure (95%, confidence intervals; 10,000 bootstrap samples), which has been recommended to test mediation effects [64] and does not require data normality to test indirect effects [65,66].

## 3. Results

### 3.1. Descriptive Statistics and Correlations

Descriptive statistics including the minimum and maximum data values, means and standard deviations for continuous variables, and absolute frequencies and percentages for categorical variables employed in this study are presented in Table 2. In addition, the rates of missing values obtained in all variables employed in this research are included in both tables. As can be observed in Table 2, mother’s and father’s occupation presented high rates of missing values (19% for the father and 13.9% for the mother), along with the global health score (20.6%), health-related quality of life (19%), and lifestyle score (29%). Contrastingly, father’s and mother’s educational level and subjective social status presented the highest completion rates (oscillating between 92.9% and 96.5%). Table 3 shows the frequency, percentages, and missing values for categories in which all variables related to lifestyles, employed to compute the scale for healthy lifestyles, were re-coded following recommended guidelines (detailed in Table 1).

Spearman correlation coefficient values, shown in Table 4, indicate that all of the socioeconomic indicators (individual objective indicators, IOSEP, and SSS) have positive and significant relationships (*p* < 0.001) with the global health score, albeit low correlations. Association between all objective socioeconomic indicators was also significant (*p* < 0.001) and moderate, except in the case of the educational level of both parents, which was highly correlated (*r* = 0.541). The SSS shows significant (*p* < 0.001) but low associations with all objective socioeconomic indicators, with family affluence presenting the highest correlation coefficient (*r* = 0.29) and mother’s education the lowest (*r* = 0.06). The healthy lifestyle score had a moderate association with health (*r* = 0.39), significantly associated (*p* < 0.001) with all socioeconomic indicators, and also showed correlation coefficients oscillating between 0.07 with SSS and 0.23 with IOSEP.

### 3.2. Measurement Model

The hypothetical measurement models were tested through a series of structural equations models. Table 5 presents the absolute fit indicator Chi-square and the approximate goodness-of-fit indices for each model. Table 6 presents the standardized path coefficients in all models. As can be observed, Chi-square was significant in all models, suggesting an inadequate model fit. However, it has been demonstrated that large sample sizes tend to increase the Chi-square value [63], as is the case in this study. For this reason, other fit indices were simultaneously considered (CFI, TLI, RMSEA, and SRMS). 

#### 3.2.1. Models Including Direct and Indirect Paths of SEP Indicators on Health

Regarding the models, including the effect of the socioeconomic status on health (see Models 1–3 and 3b in Table 5 and Table 6), data showed that Model 1, including only the predictive capacity of IOSEP, yielded a good data fit (CFI = 0.981, TLI = 0.971, RMSEA = 0.029, SRMR = 0.019) with an estimated standardized parameter of 0.265 (SE = 0.014; *p* < 0.001). Model 2 also showed good data fit (CFI = 0.990, TLI = 0.981, RMSEA = 0.027, SRMR = 0.011), with the estimated standardized parameter of SSS on health presenting a value of 0.217 (SE = 0.012; *p* < 0.001). Finally, Model 3, including both IOSEP and SSS as predictors of health, showed an acceptable data fit (CFI = 0.913, RMSEA = 0.057, SRMR = 0.005). Both estimated standardized parameters from the IOSEP and SSS on health were significant (*p* < 0.001), showing values of 0.215 (SE = 0.014) and 0.182 (SE = 0.013), respectively. However, the TLI value (0.878) indicated that model fit could be improved. Thus, considering the results of the LM test, Model 3 was re-specified to include a path between IOSEP and SSS. From a theoretical point of view, it is reasonable that the family’s objective socioeconomic conditions influence SSS. Re-specification of the structural model (see Model 3b in Table 5) yielded good model fit (CFI = 0.960, TLI = 0.941; RMSEA = 0.039, SRMR = 0.026), resulting in a significant improvement of the previous model. The decrease in Chi-square between both models was significant (*χ*² _(difference)_ = 752.14, *p* < 0.001) and the increase in CFI was higher than 0.01 (ΔCFI = 0.05). Table 6 presents an overview of the patterns of the total, direct, and indirect effects of objective SEP on adolescent health in the single mediator model. The test of the indirect effects and bootstrapped confidence interval revealed that the indirect effects of SEP on health through SSS were significant (*β* = 0.047; SE = 0.005; CI: 0.038, 0.056; *p* < 0.001).

In addition, the proportion of explained variance for each model is shown in Table 6. The model including IOSEP as a predictor presented a higher proportion of explained variance of the dependent variable (health of the adolescent population) (*R*^2^_Model 1_ = 0.070) than the model including SSS as a single predictor (*R*^2^_Model 2_ = 0.047). Model 3, including both IOSEP and SSS, showed a higher proportion of explained health than the models including only one predictor. This proportion increased even more in the re-specified Model 3b; this included the indirect effect of IOSEP on health through SSS (*R*^2^_Model 3_ = 0.079; *R*^2^_Model 3b_ = 0.092). 

#### 3.2.2. Models Including Direct Paths of SEP Indicators on Healthy Lifestyles

Regarding the models including the effect of SEP on healthy lifestyles (see Models 4–6 in Table 5 and Table 6), data showed that Model 4, including only IOSEP as predictor of healthy lifestyles, presented a good fit to the data (CFI = 0.979, TLI = 0.954, RMSEA = 0.046, SRMR = 0.021) with an estimated standardized parameter of 0.283 (SE = 0.012; *p* < 0.001). Fit indices of Model 5 are not reported given that it was a saturated model with zero degrees of freedom. However, Model 5 yielded an estimate of the standardized parameter of SSS on healthy lifestyles of 0.062 (SE = 0.010; *p* < 0.001). Finally, the estimation of Model 6, including the direct effects of IOSEP and SSS on healthy lifestyles, showed an inadequate model fit (CFI = 0.873, TLI = 0.778; RMSEA = 0.091, SRMR = 0.065), and whereas the standardized parameter of IOSEP on healthy lifestyles was significant (*β* = 0.281; SE = 0.013; *p* < 0.001), the standardized parameter of SSS on healthy lifestyles was not significant (*β* = 0.013; SE = 0.011; *p* = 0.214). In addition, the model including IOSEP as a single predictor of healthy lifestyles presented a higher proportion of explained variance (*R*^2^_Model 4_ = 0.080) than the models including only SSS (*R*^2^_Model 5_ = 0.004) and the model including the effect of both predictors, IOSEP and SSS, which revealed a similar proportion (*R*^2^_Model 6_ = 0.079).

#### 3.2.3. Model Including the Effect of the IOSEP on Health Directly and Indirectly through SSS and Lifestyles

The previous analysis showed better health prediction when the models included the effects of both IOSEP and SSS, and especially when including the mediation effect of SSS (see Models 1–3b in Table 5 and Table 6). In addition, healthy lifestyles were highly correlated with health (see Table 4) and were predicted principally by IOSEP (see Models 4–6 in Table 5 and Table 6). Thus, a final model was built including both the direct and indirect effects of IOSEP on health through SSS and healthy lifestyles. As can be observed in Table 5 (see Model 7), values of fit indices indicate an excellent model fit (CFI = 0.957, TLI = 0.940; RMSEA = 0.038, SRMR = 0.027). In addition, the proportion of health explained by Model 7 increased notably with respect to the previous models (*R^2^*_Model 7_ = 0.287). Figure 2 shows a graphic representation of the final structural model and the parameters obtained.

#### 3.2.4. Analysis of the Indirect Effects

The total, direct and indirect effects of objective SEP on adolescent health in the multiple-mediator model were examined (see Model 7 in Table 6). The indirect effects and bootstrapped confidence interval tests showed SEP to have a significant indirect effect on health through SSS (*β* = 0.049; SE = 0.004; CI: 0.040, 0.058; *p* < 0.001) and healthy lifestyles (*β* = 1.125; SE = 0.006; CI: 0.113, 0.137; *p* < 0.001). In addition, the standardized parameter for the direct path between IOSEP and health was still significant after considering the two mediation effects (*β* = 0.091; SE = 0.015; *p* < 0.001) but decreased when including the indirect effects (see the estimated standardized parameters of this path in Models 1–3b).

## 4. Discussion

The impact of inequalities on adolescent health, as well as the mechanisms through which these inequalities act, are unclear. Our research, using a wide range of objective and subjective SEP indicators and global scores for health and lifestyles, aimed to clarify how the combination of different SEP dimensions influence health, taking into account their direct and indirect effects and their influence on health through lifestyles.

The low correlation between the distinct socioeconomic indicators employed in this research (*r* = between 0.11 to 0.54) illustrates that they measure different concepts [18]. This association, especially low among objective indicators and SSS, is explained in a recent qualitative study conducted by Martin-Storey et al. [67] which found that adolescents tend to perceive their own status placement based on traditional SEP markers (such as money, material goods, or education), however they also take into account other aspects such as societal factors (social status, quality of relationships, etc.), or values (around work, perseverance, etc.). 

Regarding the direct effect of objective SEP on health and healthy lifestyles, the global composite index (IOSEP) showed to associate both adolescent health and lifestyles. This finding is consistent with previous research demonstrating parental educational level, parental occupational status, and family affluence to have an effect adolescent health and healthy lifestyle outcomes [10,68,69]. Moreover, these objective indicators have previously shown to better predict adolescent health when considered together rather than when used individually [17]. This research adds to the body of existing literature that defends that using a global index to evaluate adolescent SEP could also be a valid approach to examine inequalities in health-related behaviors.

When the effects of SSS were analyzed individually, results showed that perceived family wealth was a significant predictor of both adolescent health and lifestyles. With respect to health, SSS showed a significant effect even when controlling for the effect of IOSEP and health, congruent with previous research [21,22]. However, our results differ from other studies [24,25] in which SSS is reported to have a higher impact on health than objective indicators. Moreover, when comparing the models of objective and subjective SEP indicators (both individually and combined), our results show SSS to mediate the relationships between objective SEP and health. Certain circumstances of material wealth can lead one to feel economically poor compared to their peers, which in turn has negative psychological consequences such as stress or anxiety [70] and thus plays an important role in the association between SEP and health [71]. Therefore, health inequalities are more accurately estimated when employing both types of indicators and taking into account the direct effect of objective SEP and its indirect effect through perceived wealth.

Regarding healthy lifestyles, SSS showed a weak association that remained insignificant when the effect of IOSEP was considered. These results are in line with previous findings showing equalization and unstable association patterns between SSS and adolescent lifestyles [72], and a reduction in this association after controlling the effects of objective SEP [26]. However, this finding should be considered with caution. As was previously mentioned, SSS may be related to certain health behaviors, however this relationship is underestimated when healthy lifestyles are considered globally. For example, previous findings have shown a relationship between SSS and behaviors such as fruit and vegetables consumption, physical activity [73], or sleeping habits [74], even after controlling objective SEP. Moreover, an inverse relationship has also been found between SSS and behaviors such as smoking or alcohol use [56,75]. Thus, the effect of SSS on various health-related behaviors can also be counteracted by opposite associations. Despite these considerations, our findings suggest that inequalities exist in adolescent health and lifestyles, and that equalization can be an artefact due to inaccurate assessment. In addition, the impact of objective socioeconomic factors on health are increased by adding the subjective perception of wealth, as material conditions are more important for healthy lifestyles than feelings of deprivation.

Lastly, lifestyles mediated the relationships between socioeconomic position and health, as other research has found [31,76,77,78]. Moreover, the model explained 28.7% of adolescent health when the direct effect of objective SEP and its indirect effects through SSS and healthy lifestyles were considered together. Therefore, these results highlight the important role that objective socioeconomic conditions play on adolescent health, having direct and indirect effects through subjective socioeconomic status and healthy behaviors. Moreover, as previous research has found [7,79], the direct effect of objective SEP on health, despite being reduced, did not disappear when the influence of lifestyles and psychological mechanisms was controlled.

### 4.1. Methodological Considerations and Implications of this Study

Global scores used to assess adolescents’ objective SEP, health, or lifestyles do not detect weak or non-association between certain socioeconomic indicators and specific health or lifestyles outcomes. However, we believe that SEP can be better understood by employing composite indices, thus capturing the synergistic effect of its different dimensions on adolescent health [36,80]. Therefore, among the strengths of this study, it is worth mentioning that this present research applies a recently validated index for measuring adolescents’ objective socioeconomic position (IOSEP) [17], in addition to using multiple measures to assess health and healthy lifestyles. Moreover, the large representative sample that this present study uses adds to the validity of the measures. 

However, some limitations should also be mentioned. Firstly, the information was provided exclusively by the adolescents and therefore refers to their own perceptions. Secondly, this research did not employ the MacArthur scale [21,81]—the most common measure of SSS—which allows adolescents to indicate graphically where they perceive themselves to be in relationship to their peers or where their family is in relation to society. Nonetheless, a recent meta-analysis of the effect of subjective socioeconomic position on health showed similar results using the measures society ladder, school ladder, and financial constrains [72]. Thirdly, the scale used to evaluate health-related behaviors showed a low value of Cronbach’s alpha. This low correlation between health-related behaviors is the reason that they were not represented as global factors. Therefore, following previous research [82,83], we employed indices summarizing individual scores in each behavior, classified as more or less healthy, despite the scale presenting low levels of consistency [84]. Fourthly, the cross-sectional data do not allow causal one-directional relationships and mediation to be established, in the strict sense [85]. Casual relationships were assumed based on the literature, however the health of children and adolescents can influence their family’s economic status [86]. Fifthly, with respect to the use of global scores, further research should explore the invariance of the measures and the explanatory models proposed across time and space. In addition, this study did not examine the effect of other factors, such as sex, which are also sources of health inequalities. Therefore, further research should explore the invariance of the proposed models across other characteristics of the studied populations, such as sex, age, ethnicity, or even across different groups according to their objective/subjective socioeconomic status. Finally, research of economic disadvantages on adolescent health could benefit from further studies examining alternative models and include other mechanism by which socioeconomic differences influence health, such as the adolescents’ educational level, family structure, housing conditions or quality of public services, community social capital, and factors at the neighborhood level or, on a macroscopic level, such as unemployment rates or distribution of Gross Domestic Product.

## 5. Conclusions

This study highlights the potential role of multiple factors that underlie the relationship between socioeconomic status and adolescent health. Moreover, this research demonstrated that the impact of socioeconomic inequalities on adolescent health is underestimated when examining only the direct effects of material wealth. Objective socioeconomic conditions were shown to affect health both directly and indirectly through SSS and health-related behaviors. Therefore, the perception of a low socioeconomic status and the impact of socioeconomic position on adolescents’ healthy lifestyles was shown to play an important role in explaining the impact of objective socioeconomic indicators on adolescent health. Thus, interventions aimed at reducing inequalities in this developmental stage could improve their efficacy by considering the negative consequences provoked by adolescents perceiving themselves as poor with respect to a certain socioeconomic position. Consequently, designing educational and social interventions aimed at building a more egalitarian and inclusive society can be a complementary measure to programs aimed at reducing health inequalities, especially among the most disadvantaged adolescents. 

Likewise, prevention programs should target the unhealthy behaviors of adolescents from lower socioeconomic groups to help prevent future life-course disadvantages. Moreover, this research showed that objective socioeconomic conditions affect health directly and indirectly through the perception of a low socioeconomic status, however health-related behaviors seem to be more influenced by material conditions. Thus, research efforts in healthy lifestyle promotion among adolescents are beneficial for their health and can contribute to reducing health inequalities. Therefore, it is important to continue promoting healthy lifestyles (such as healthy eating or sleeping habits, or patterns of physical activity) and abstaining from of substance use behaviors, through social media and interventions at schools or in social contexts such as the neighborhoods, which focus especially on more disadvantaged adolescents. In fact, this research indicated the need to equalize adolescents’ access to healthy lifestyles by reducing material inequalities (such as facilitating the availability of healthy food or eliminating socioeconomic barriers in the access to sports and psychical activity).

Further research should continue identifying the specific socioeconomic factors influencing health and lifestyles, as well as the more susceptible health outcomes or behaviors, in order to concentrate intervention efforts on reducing its impact. In addition, it is necessary further research the mechanism that explain the effect through which inequalities exert their influence on health and how they interact. 

## Figures and Tables

**Figure 1 ijerph-16-01637-f001:**
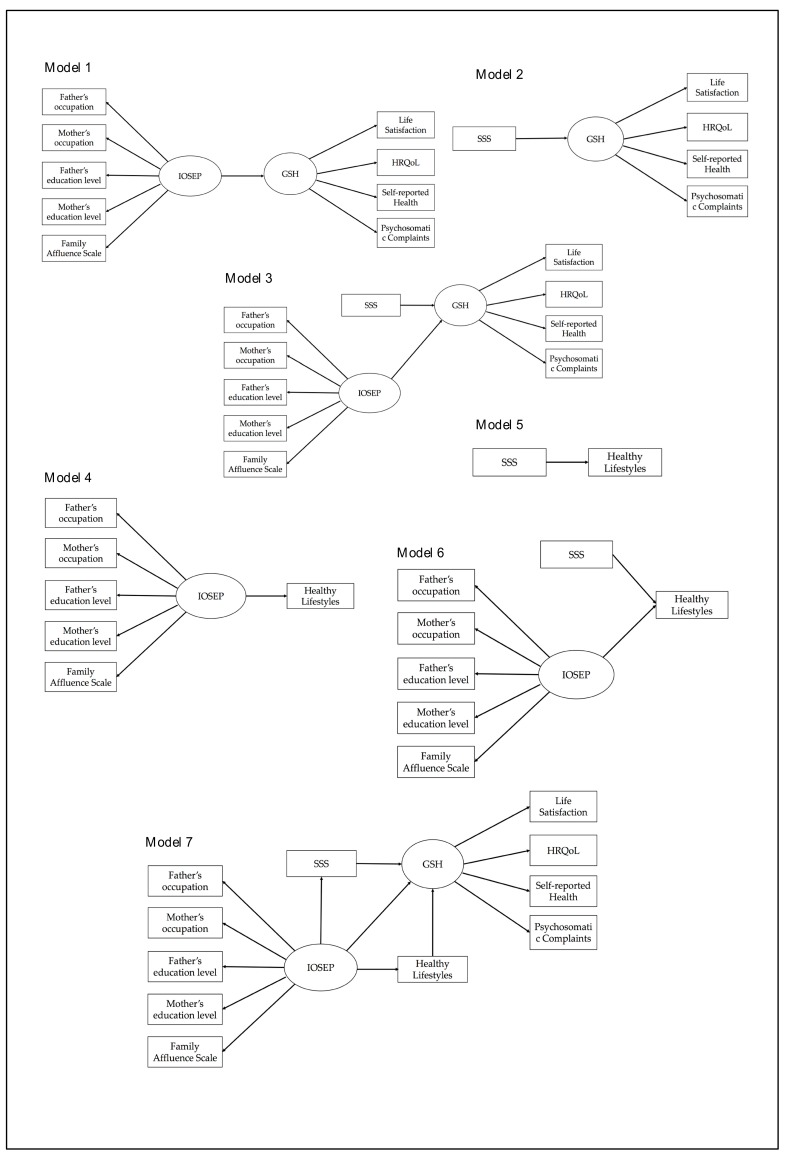
The hypothesized models. IOSEP, index of objective socioeconomic position; SSS, subjective socioeconomic status; GHS, global health score; HRQoL, health-related quality of life.

**Figure 2 ijerph-16-01637-f002:**
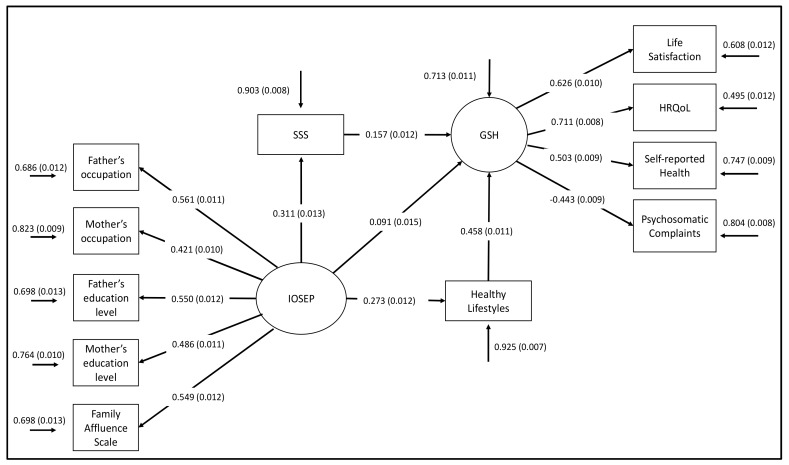
Path coefficient estimates of the final model (Model 7). Note: IOSEP, index of objective SEP; SSS, subjective socioeconomic status; GSH, global score of health; HRQoL, health-related quality of life. The error terms of the variables maternal and paternal education, and those from the variables maternal education and maternal occupation were allowed to covary, as was previously demonstrated in Moreno-Maldonado, Rivera, Ramos, and Moreno [16]. The error term of the IOSEP was not estimated, being fixed to 1. *p* ˂ 0.05.

**Table 1 ijerph-16-01637-t001:** Selected variables and the instruments used for their assessment in the present study.

**Measures to Assess Socioeconomic Status**	
Objective Socioeconomic Status. It was assessed using IOSEP (Index of Objective Socioeconomic Position), comprised of:	
Education level of both parents	Scored on 4 levels, parental educational level was evaluated by the question “What level of education do your father and mother have?” The response options for the question referring to each parent were: “*Never studied (does not know how to read or write, or does so with difficulty)*”;“*Basic/primary studies finished or unfinished*” (specifying: “*Something similar to what I am studying now*” for the 11–12 year-old participants); “*Secondary studies (like high-school or vocational school), finished or unfinished*” (specifying: “*They studied at a higher level than what I am currently studying*” for the 11–12 year-old participants); “*University studies, finished or unfinished*”. The four education-levels corresponded with the *International Standard Classification of Education* (ISCED) adopted by the UNESCO General Conference in 1997 [41]: ISCED 0; ISCED 1-2; ISCED 3-4; and ISCED 5-6.
Occupational status of both parents	Classified in the 10 categories proposed by the *International Standard Classification of Occupations* (ISCO-08) at the highest level of aggregation [42], and adding an extra category to include unemployment.
*Family Affluence Scale* (FAS)	FAS was used to measure family material wealth through the six items that make up the latest version of the instrument [43]: Number of family cars or family computers, own bedrooms, number of bathrooms at home, family dishwasher, and family holidays abroad. Alpha reliability for the 6-item scale was 0.96, indicating excellent internal consistency.
Subjective Socioeconomic Status	
Perceived family wealth	It was assessed by the question: “How well-off do you think your family is?” The question has been used in the HBSC study since 1994 as an indicator of the adolescents’ subjective socioeconomic status. The 5 response options were classified in 4 categories: 1 *(poor)*, 2 *(not very poor)*, 3 *(normal)*, and 4 *(rich or very rich)*.
**Measures to Evaluate Health**	
GHS: Global Health Score. Composed of:	
Life satisfaction	Evaluated through the instrument “Cantril Ladder Scale” [44], which ranges from 0 *the lowest perception of life satisfaction* to 4 *the highest*.
Heath-related quality of life	Assessed through the instrument “Kidscreen”, consisting of 10 items that evaluate aspects of physical, psychological, and social health in a Likert scale of 5 points [45]. Alpha reliability for the 10-item scale was 0.83.
Self-reported health	Adolescents were asked how they rated their health with four response options: excellent, good, passable, or poor [46,47].
Psychosomatic complaints	An HBSC-symptom checklist [48] was employed asking adolescents how often in the last six months they had experienced certain symptoms. Response options for each symptom ranged from “almost every day” to “rarely or never”, and the maximum frequency of experiencing any psychological or somatic symptom was calculated. The 8-item scale showed an alpha Combrach of 0.83 indicating good internal consistency.
**Measures to Evaluate Healthy Lifestyles**	
Global score of healthy lifestyles	
Frequency of breakfast	The adolescents were asked: “How often do you have breakfast (something more than a glass of milk or fruit juice)?” The responses were classified from 0—infrequently (*never or almost never*), 1—irregular (*from 2 to 6 days a week*), 2—daily (*7 days a week*).
Eating habits	The specific question was: “How many times a week do you eat/drink fruit/vegetables/sweets/sugary soft drinks?” The response categories collected information about the weekly consumption frequency of the cited products, classifying the responses in three categories. In the case of fruits and vegetables: 0—infrequently (*never or less than once a week*), 1—irregularly (*from 1 to 6 times a week*), 2—daily (*every day, and every day more than once a day*). In the case of sweets and soft drinks, the responses were classified in the same way but inversely, with daily consumption being the least healthy (0). Current scientific evidence [49] was taken into consideration for the categorization of the responses as more or less healthy in the case of frequency of breakfast and dietary habits.
Tooth brushing	Evaluated with the question “How often do you brush your teeth?” The responses were classified in the following categories: 0—irregularly or never (*less than once a day*), 1—daily (*once a day*), 2—optimal (*more than once a day*). At least twice a day is an indicator of a universally accepted healthy lifestyle [50].
Physical activity	Physical activity was evaluated by two questions adapted for their use in the HBSC [51]: (1) number of days over the past 7 days in which the adolescents were physically active for at least 60 minutes a day (moderate-to-vigorous physical activity), classified in three categories: 0—infrequently (*less than two days a week*), 1—irregularly (*from 3 to 4 days a week*), 2—regularly (*5 days or more a week*); (2) frequency that adolescents performed some physical activity in their free time that made them sweat or out of breath (vigorous physical activity), which was classified in three categories: 0—infrequently (*never or less than once a month*), 1—irregularly (*once a month or once a week*), 2—regularly (*two days a week or more*). The classification of more- or less-healthy behavior was based on recommendations established in scientific literature [52,53].
Hours of sleep	An average score of the number of hours a day that adolescents sleep was calculated and classified in 3 groups according to criteria establishing a minimum of 8-hours of sleep for optimal rest in adolescence [54]. The responses were classified in 3 groups: 0—insufficient (*less than 6.5 hours*), 1—sufficient (*between 6.5 and 7.5 hours*) and 2—optimal rest (*at least 8 hours of sleep*).
Tobacco use	Assessed through the question: “How often do you smoke tobacco at present?” with 4 response options: “every day”, “smoke, at least once a week, but not every day”, “less than once a week” and “never smoked”. The responses were classified into three groups: 0—daily smokers, 1—experimenters (smoke, at least once a week, but not every day or less than once a week), and 2—never smoked. This classification has been previously employed [55].
Alcohol consumption	Assessed by asking about their frequency of drinking beer, wine/sparkling wine, spirits/liquor, alcopops, and other alcoholic beverages, with 5 response options for each item: every day, every week, every month, rarely, and never. The maximum frequency of alcohol consumption was calculated independently of the type of alcoholic beverage, and were re-coded in 0—regular users (*used at least one of the alcoholic beverages every week or more often*); 1—irregular users (*drank alcohol every month or every week*); and 3—not users of alcohol (*answered never for all alcoholic beverages*). These categories have been used in previous HBSC studies [56].

**Table 2 ijerph-16-01637-t002:** Descriptive characteristics of the sample population in all variables analyzed (*n* = 15,340 adolescents aged 11–16 years old).

	Mean	SD	Min. Max	*n*	%	% Missing
Index of Objective Socioeconomic Position	0.04	1.73	(−5.32, 4.32)			35.3
Father’s occupational status			(1, 10)			19.0
Unemployed				1640	10.7	
Elementary Occupations				492	3.2	
Plant and Machine Operators Assemblers				1500	9.8	
Craft and Related Trades Workers				2400	15.6	
Skilled Agricultural, Forestry and Fishery Workers				873	5.7	
Services and Sales Workers				1817	11.8	
Clerical Support Workers				414	2.7	
Technicians and Associate Professionals				1135	7.4	
Professionals				1322	8.6	
Managers				829	5.4	
Mother’s occupation			(1, 10)			13.9
Unemployed				4478	29.2	
Elementary Occupations				1410	9.2	
Plant and Machine Operators Assemblers				203	1.3	
Craft and Related Trades Workers				405	2.6	
Skilled Agricultural, Forestry and Fishery Workers				149	1.0	
Services and Sales Workers				2978	19.4	
Clerical Support Workers				886	5.8	
Technicians and Associate Professionals				710	4.6	
Professionals				1703	11.1	
Managers				291	1.9	
Father’s educational level			(1, 4)			6.4
Pre-primary education				232	1.5	
Basic education				4914	32.0	
Secondary education				5466	35.6	
Higher education				3752	24.5	
Mother’s educational level			(1, 4)			3.5
Pre-primary education				257	1.7	
Basic education				4390	28.6	
Secondary education				5364	35.0	
Higher education				4797	31.3	
Family Affluence Scale	9.13	2.15	(1, 14)			9.4
Subjective Socioeconomic Status			(1, 5)			7.1
Poor				121	0.8	
Very poor				812	5.3	
Normal				1037	6.8	
Rich				340	2.2	
Very rich				1377	9	
Global Health Score	0.13	0.97	(−5.02, 2.69)			20.6
Life satisfaction	8.78	2.02	(1, 11)			10.2
Health-related quality of life	35.22	4.56	(10, 50)			19.0
Self-reported health			(1, 4)			9.8
Poor				116	0.8	
Passable				918	6.0	
Good				6963	45.4	
Excellent				5834	38.0	
Psychosomatic complaints			(1, 5)			9.8
Rarely				2310	15.1	
Never				2957	19.3	
Often				1938	12.6	
About every week				2921	19.0	
About every day				3717	24.2	
Healthy lifestyles score	15.94	2.81	(3, 22)			29.00

SD = standard deviation.

**Table 3 ijerph-16-01637-t003:** Descriptive characteristics of the sample population (*n* = 15,340) in all health behaviors used to calculate the global health score.

	*n*	%	% Missing
Breakfast consumption			8.7
Infrequently	663	4.3	
Irregular	3634	23.7	
Daily	9705	63.3	
Fruit consumption			2.1
Infrequently	1835	12.0	
Irregular	7954	51.9	
Daily	5229	34.1	
Vegetables consumption			3.1
Infrequently	1598	10.4	
Irregular	10,089	65.8	
Daily	3180	20.7	
Sweets consumption			2.8
Daily	1977	12.9	
Irregular	8673	56.5	
Infrequently	4264	27.8	
Sugary-drinks consumption			3.1
Daily	3120	20.3	
Irregular	7826	51.0	
Infrequently	3913	25.5	
Tooth brushing			1.9
Irregularly/never	1144	7.5	
Frequent	3863	25.2	
Optimal	10,045	65.5	
Moderate-to-vigorous physical activity			5.1
Irregular	2655	17.3	
Infrequently	4468	29.1	
Optimal	7434	48.5	
Vigorous physical activity			14.2
Infrequently	1627	10.6	
Irregular	2018	13.2	
Optimal	9516	62.0	
Hours of sleep			3.3
Insufficient	637	4.2	
Sufficient	2269	14.8	
Optimal rest	11,921	77.7	
Tobacco use			3.3
Daily	435	2.8	
Experimental	722	4.7	
Never	13,672	89.1	
Alcohol consumption			2.1
Weekly	735	4.8	
Infrequent	5110	33.3	
Never	9172	59.8	

Categorization of each behavior as more or less healthy was based on international guidelines (described in Table 2).

**Table 4 ijerph-16-01637-t004:** Spearman’s correlation coefficients (rho) between socioeconomic indicators, adolescent health, and healthy lifestyles.

		Single Socioeconomic Objective Indicators					SSS	GHS	Single Health Indicators				HLS
		1.1	1.2	1.3	1.4	1.5	2	3	3.1	3.2	3.3	3.4	4
1	IOSEP	0.66	0.57	0.74	0.72	0.60	0.24	0.16	0.13	0.12	0.12	−0.08	0.23
1.1	OCC F		0.22	0.36	0.26	0.28	0.16	0.09	0.08	0.06	0.07	−0.05	0.12
1.2	OCC M			0.23	0.38	0.24	0.11	0.06	0.05	0.03	0.05	−0.05	0.10
1.3	EDL F				0.54	0.24	0.12	0.12	0.08	0.10	0.089	−0.07	0.20
1.4	EDL M					0.27	0.12	0.13	0.09	0.11	0.10	−0.07	0.21
1.5	FAS						0.29	0.15	0.14	0.11	0.11	−0.06	0.12
2	SSS							0.18	0.18	0.13	0.14	−0.09	0.07
3	GHS								0.78	0.84	0.60	−0.51	0.39
3.1	LS									0.50	0.32	−0.29	0.29
3.2	HRQoL										0.38	−0.29	0.35
3.3	SRH											−0.22	0.24
3.4	PSC												−0.24

All correlations were significant at the 0.001 level. IOSEP, index of objective socioeconomic position; OCC F, father’s occupation; OCC M, mother’s occupation; EDL F, father’s educational level; EDL M, mother’s educational level; FAS, family affluence scale; SSS, subjective socioeconomic status; GHS, global health score; LS, life satisfaction; HRQoL, health-related quality of life; SRH, self-reported health; PSC, psychosomatic complaints; HLS, healthy lifestyles.

**Table 5 ijerph-16-01637-t005:** Goodness-of-fit indices for all proposed models.

	Model 1	Model 2	Model 3	Model 3b ^a^	Model 4 ^b^	Model 6	Model 7
χ²	343.92	55.62	1489.99	737.86	236.92	1424.03	883.14
*p*	<0.001	<0.001	<0.001	<0.001	<0.001	<0.001	<0.001
gl	24	5	32	31	7	12	39
CFI	0.981	0.990	0.913	0.960	0.979	0.873	0.967
TLI	0.971	0.981	0.878	0.941	0.954	0.778	0.940
RMSA (CI 90%)	0.029 (0.027, 0.032)	0.027 (0.021, 0.034)	0.057 (0.054, 0.059)	0.039 (0.036, 0.041)	0.046 (0.041, 0.051)	0.091 (0.087, 0.095)	0.038 (0.035, 0.040)
SRMR	0.019	0.011	0,005	0.026	0.021	0.065	0.027

^a^ Model 3b is Model 3 re-specified, including the path from the index of objective socioeconomic status and subjective social status, suggested by the LM Test and based on theoretical evidence. ^b^ Goodness-of-fit indices for Model 5 are not presented because it was a saturated model with zero degrees of freedom. CFI, Comparative Fit Index; TLI, Tucker–Lewis Index; RMSA, Root Mean Square Error of Approximation; CI, confidence intervals; SRMR, Standardized Root Mean Squared Residual.

**Table 6 ijerph-16-01637-t006:** Standardized coefficients representing direct and indirect paths for the models.

Effects	B	SE	*p*	95% CI
Models including only direct effects on health				
IOSEP (Model 1)	0.265	0.014	<0.001	[0.238; 0.210]
*R^2^*	0.070			
SSS (Model 2)	0.217	0.012	<0.001	[0.193; 0.241]
*R^2^*	0.047			
IOSEP (Model 3)	0.215	0.014	<0.001	[0.187; 0.242]
SSS	0.182	0.013	<0.001	[0.157; 0.207]
*R^2^*	0.079			
Models including only direct effects on healthy lifestyles				
IOSEP (Model 4)	0.283	0.012	<0.001	[0.259; 0.307]
*R^2^*	0.080			
SSS (Model 5)	0.062	0.010	<0.001	[0.041; 0.082]
*R^2^*	0.004			
IOSEP (Model 6)	0.281	0.013	<0.001	[0.256; 0.305]
SSS	0.013	0.011	*ns*, 0.214	[−0.008; 0.034]
*R^2^*	0.079			
Single mediator model (Model 3b)				
IOSEP to mediator (SSS)	0.321	0.013	<0.001	[0.296; 0.347]
Mediator (SSS) to health	0.146	0.013	<0.001	[0.120; 0.172]
Total effect of IOSEP on health	0.271	0.014	<0.001	[0.244; 0.298]
Direct effect of IOSEP on health	0.224	0.015	<0.001	[0.195; 0.253]
Total indirect effect of IOSEP on health through SSS	0.047	0.005	<0.001	[0.038; 0.056]
*R^2^*	0.092			
Multiple mediator model (Model 7)				
IOSEP to mediators				
SSS	0.311	0.013	<0.001	[0.286; 0.337]
Healthy lifestyles	0.273	0.012	<0.001	[0.250; 0.297]
Mediator to health				
SSS	0.157	0.012	<0.001	[0.133; 0.182]
Healthy lifestyles	0.458	0.011	<0.001	[0.436; 0.480]
Total effect of IOSEP on health	0.265	0.013	<0.001	[0.239; 0.291]
Direct effect of IOSEP on health	0.091	0.015	<0.001	[0.062; 0.120]
Total indirect effect of IOSEP on health through mediators	0.174	0.007	<0.001	[0.160; 0.188]
SSS	0.049	0.004	<0.001	[0.040; 0.058]
Healthy lifestyles	0.125	0.006	<0.001	[0.113; 0.137]
*R^2^*	0.287			

95% Confidence intervals (95% CIs) for the indirect effects are based on bootstrapping method. IOSEP, index of objective socioeconomic position; SSS, subjective socioeconomic status.

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
