# Peer review of "Direct and Indirect Influences of Objective Socioeconomic Position on Adolescent Health: The Mediating Roles of Subjective Socioeconomic Status and Lifestyles"

_ijerph, 2019, doi:10.3390/ijerph16091637_

Round 1

Reviewer 1 Report

This is an interesting study that examines the associations between socioeconomic position and youth health outcomes among youths in Spain. Below, I address a few issues/questions that emerged from my read of the paper.

Introduction

Author(s) addressed several interesting multiple direct and indirect association hypotheses between socioeconomic position and youth health (shown in Figure 1). To describe each hypothesis, the author(s) briefly mentioned some of the inconsistent previous findings. However, the mechanism for each model was not directly described in the intro. Without a clear explanation of these processes, it is hard to follow the hypotheses. If there is no space to describe all hypotheses in the introduction, I think that author(s) may directly start with model 3 because model 3 includes two previous model perspectives (models 1 and 2).

In terms of two socioeconomic position perspectives (i.e., objective and subjective socioeconomic positions), author(s) addressed the effects of both objective and subjective socioeconomic status on youth health outcomes (e.g., model 3 of figure 1). However, I think there may be potential multiplicative effects (i.e., interaction effects) between objective and subjective socioeconomic status on health outcomes. For example, there may be some distinct groups such as both objectives and subjective high group, only objective high, only subjective high, both low group. When I saw the correlation matrix (shown in table 4), most objective socioeconomic indicators has small-to-moderated associations with a subjective socioeconomic indicator, implying the existences of distinct groups between subjective and objective socioeconomic groups. I assume that each distinct socioeconomic position group may have unique direct and indirect processes on youth health outcomes through lifestyle.  

In addition, the construct of a healthy lifestyle is questionable. It seems that this lifestyle in adolescence is essential to understand the socioeconomic processes on health outcomes. Author(s) briefly mentioned in several places about the previous inconsistent findings. However, before addressing these inconsistent findings, I think that it is necessary to address why and how this lifestyle can be one of the key mediators between socioeconomic position and health outcomes.

Also, the construct of lifestyle is questionable. In addition, when I read the measures section, I noticed that this lifestyle was created by a series of individual health behaviors. To improve readers’ understanding, I suggest briefly introducing why summed scores of multiple health behaviors are important (instead of the latent construct).

Data analytical plan

Author(s) mentioned that the Maximum likelihood with robust standard errors was used to estimate coefficients because of non-normal data. Please clarify which variables cannot hold normality.

Previous health research has indicated that there are significant gender and race/ethnicity differences in health-related behaviors. Please consider gender and races/ethnicity as control variables.  

Discussion

Given that the author(s) used cross-sectional data, the term “predict” which author(s) used may make less sense (on line 304 of page 13). I suggest using the term “associate” instead of “predict”.

The practical implication could be much stronger. It would be useful to consider how the specific findings in this study inform routine clinical care and interventions for Spain youth.

Author Response

Dear Editor and reviewers,

First of all we would like to express our gratitude for accepting to review this manuscript and for the insightful review of the earlier version, which we have found very helpful to improve the study. Thus, we are submitting a revised version of the manuscript that has been enriched based on the comments, to each of which we have responded. All changes in the manuscript are highlighted in yellow. Reviewer’s comments are shown below and our responses are indicated in italics after each of them.

First of all, we address comments of Reviewer #1.

This is an interesting study that examines the associations between socioeconomic position and youth health outcomes among youths in Spain. Below, I address a few issues/questions that emerged from my read of the paper.

 Introduction

Author(s) addressed several interesting multiple direct and indirect association hypotheses between socioeconomic position and youth health (shown in Figure 1). To describe each hypothesis, the author(s) briefly mentioned some of the inconsistent previous findings. However, the mechanism for each model was not directly described in the intro. Without a clear explanation of these processes, it is hard to follow the hypotheses. If there is no space to describe all hypotheses in the introduction, I think that author(s) may directly start with model 3 because model 3 includes two previous model perspectives (models 1 and 2).

-       We thank the reviewer for the comment. We have completed the introduction section with the conceptual model of the Commission on Social Determinants on Health [1], explaining in more detail how socioeconomic position influences health through material, psychological and behavioral mechanisms. In addition, we have also described the hypothesized models according to the theoretical background. We agree with the reviewer that the introduction section has improved with a more detailed description of the mechanisms through which socioeconomic position exert its influence on health and more clearly re-defining the hypothesis.

 In terms of two socioeconomic position perspectives (i.e., objective and subjective socioeconomic positions), author(s) addressed the effects of both objective and subjective socioeconomic status on youth health outcomes (e.g., model 3 of figure 1). However, I think there may be potential multiplicative effects (i.e., interaction effects) between objective and subjective socioeconomic status on health outcomes. For example, there may be some distinct groups such as both objectives and subjective high group, only objective high, only subjective high, both low group. When I saw the correlation matrix (shown in table 4), most objective socioeconomic indicators has small-to-moderated associations with a subjective socioeconomic indicator, implying the existences of distinct groups between subjective and objective socioeconomic groups. I assume that each distinct socioeconomic position group may have unique direct and indirect processes on youth health outcomes through lifestyle.  

-       There is no doubt that interaction effects between objective and subjective socioeconomic status will be also an interesting approach. Specifically, our research team is also analyzing health inequalities from this approach. Moreover, we are also interested in exploring (with a mixed approach of both, quantitative and qualitative methods) how adolescents build their socioeconomic status and factors that are on  based on their social comparisons. Nonetheless, we consider this topic to be suitable for other research, given that the principal aim of the present study is to explore mediation effects. However, we have included in the limitations section and indications for further research the need to test the invariance of the different proposed models across socioeconomic position, as well as by gender, age or ethnicity (as proposed in another comment), which we also consider to be a necessary step for generalizing the results obtained.

In addition, the construct of a healthy lifestyle is questionable. It seems that this lifestyle in adolescence is essential to understand the socioeconomic processes on health outcomes. Author(s) briefly mentioned in several places about the previous inconsistent findings. However, before addressing these inconsistent findings, I think that it is necessary to address why and how this lifestyle can be one of the key mediators between socioeconomic position and health outcomes.

-       As mentioned before, we have enriched the introduction section by better explaining the role of the behavioral factors explaining health inequalities. Moreover, we have completed the introduction section with respect to the definition and components of lifestyles and its role during adolescence.

Also, the construct of lifestyle is questionable. In addition, when I read the measures section, I noticed that this lifestyle was created by a series of individual health behaviors. To improve readers’ understanding, I suggest briefly introducing why summed scores of multiple health behaviors are important (instead of the latent construct).

-       We agree that the rationale for employing a summed score for evaluating adolescent lifestyles wasn’t clear, and this question was only briefly addressed in limitations when discussing the questionable value of the Cronbach’s Alpha. Thus, we have completed information about the conceptualization of the global score of adolescent healthy lifestyles in the instruments section.

In this regards, no latent factor was drawn due to the lack of unidimensionality among healthy lifestyles. Based on the multidimensional nature of health related behaviours, we didn’t attempt to create a factor for healthy lifestyles. However, we created a scale based on a wide range of behaviours related to lifestyles that have been demonstrated to be key for adolescent health: frequency of breakfast, dietary habits (consumption of fruit, vegetables, sweets and soft drinks) , tooth brushing, physical activity (moderate-to-vigorous physical activity and vigorous physical activity), hours of sleep, tobacco use and alcohol consumption. More studies have employed indices summarizing individual scores on health behaviours, classified as more or less healthy as was done in this research [2-5], despite the scale presenting low levels of consistency [6].

 Data analytical plan

Author(s) mentioned that the Maximum likelihood with robust standard errors was used to estimate coefficients because of non-normal data. Please clarify which variables cannot hold normality.

-       Due to the variables’ characteristics (some of them were not continuous, as the Likert scales) and the non-normality of some analyzed indicators (e.g. Education level of both parents; Occupational status of both parents, and Perceived family wealth), it is recommended the use of robust estimators.

Previous health research has indicated that there are significant gender and race/ethnicity differences in health-related behaviors. Please consider gender and races/ethnicity as control variables.  

-       We thank the reviewer for the comment, and also agree with the need and the interest of controlling, and even exploring, interaction effects between different inequalities generated not only by socioeconomic factors, but also by gender and ethnicity. However, the complexity of the proposed models and statistical analysis performed made it difficult to include more variables in the examined models and the authors think that this type of analysis will make the manuscript more complex and more difficult for the reader to understand the main conclusions of this research. However, we agree and discussed during the statistical analyses stage that an interesting approach for future research will be testing the invariance of the examined models across different factors such as gender or ethnicity, as well as for age or even in different groups according to their objective/subjective socioeconomic status. Therefore, we have enriched the limitations section in the manuscript, and propose the invariance analysis for further research.

 Discussion

Given that the author(s) used cross-sectional data, the term “predict” which author(s) used may make less sense (on line 304 of page 13). I suggest using the term “associate” instead of “predict”.

-       Thanks for the suggestion. This word has been changed.

The practical implication could be much stronger. It would be useful to consider how the specific findings in this study inform routine clinical care and interventions for Spain youth.

-       We have improved the implication section of this manuscript reflecting more about how our results can offer guidelines to improve interventions aimed at promoting health and reducing economic disparities. We thank you for the suggestion.

We hope that all the changes described have contributed to address the reviewers’ comments, in the sense that implications and relevance of the study can now more easily be judged thanks to the explicit statement of research questions and the improvements made specially in the introduction section.

To conclude, the bibliographic references used in this document are included below:

1.       Solar, O.; & Irwin, A. A conceptual framework for action on the social determinants of health (Social Determinants of Health Discussion Paper 2). World Health Organization: Geneva, Switzerland, 2010. Retrieved from: http://apps.who.int/iris/bitstream/10665/44489/1/9789241500852_eng.pdf (Access on May 20, 2018).

2.       Senn, T.E.; Walsh, J.L.; Carey, M.P. The mediating roles of perceived stress and health behaviors in the relation between objective, subjective, and neighborhood socioeconomic status and perceived health. Ann Behav Med 2014, 48, 215-24. doi:10.1007/s12160-014-9591-1.

3.       Walsh, J.L.; Senn, T.E.; Carey, M.P. Longitudinal associations between health behaviors and mental health in low-income adults. Transl Behav Med 2013, 3, 104-13. doi:10.1007/s13142-012-0189-5.

4.       Heinrich, K.M.; Maddock, J.; Multiple health behaviors in an ethnically diverse sample of adults with risk factors for cardiovascular disease. Perm J 2011, 15, 12-8. 

5.       de Vries, H.; van 't Riet, J.; Spigt, M.; Metsemakers, J.; van den Akker, M.; Vermunt, J.K. et al. Clusters of lifestyle behaviors: results from the Dutch SMILE study. Prev Med 2008, 46, 203-8. doi:10.1016/j.ypmed.2007.08.005.

6.       Donovan, J.E.; Jessor, R.; Costa, F.M. Adolescent health behavior and conventionality-unconventionality: An extension of problem-behavior therapy. Health Psychol 1991, 10, 52-61. doi: 10.1037//0278-6133.10.1.52.

Reviewer 2 Report

From my point of view the most interesting parts of this paper are: global indexes of a adolescent health and using a structural modeling to explain the connections between social posiotion and health.
I suggest the following changes:
a) to add some changes in the Introduction:
Generally I suggest to change introduction and write more about adolescents lifestyle and health. I suggest to add a lifestyle definition and its components in adolescents.  I see a lack of balance between descriptions of socioeconomic position and health and healthy lifestyle. I suggest to begin with some general information concerning adolescents health determinants and after that explain why a socioeconomic position is so important. I suggest to use one of the ecological theory of health or development as a theoretical becground. It will be easier to understand that the socioeconomic posiotion could be one of the direct or indirect aspects which influence on adolescents health. 
1. line 85 - to add a separate paragraf with the aims (subtitle)

b) to add some changes in the Matherials and Methods
1. line 123 - Authors should explain why it is better to use GHS than separate dimensions.
2. line 130 to add Global score of healthy lifestyle after Thus,

c)  in the Discusion - I suggest to write more about the limitations of this paper. I suggest a separate paragraf on the end of the paper.

Author Response

Dear Editor and reviewers,

First of all we would like to express our gratitude for accepting to review this manuscript and for the insightful review of the earlier version, which we have found very helpful to improve the study. Thus, we are submitting a revised version of the manuscript that has been enriched based on the comments, to each of which we have responded. All changes in the manuscript are highlighted in yellow. Reviewer’s comments are shown below and our responses are indicated in italics after each of them.

Addressing Reviewer 2 reports:

From my point of view the most interesting parts of this paper are: global indexes of a adolescent health and using a structural modeling to explain the connections between social posiotion and health.

I suggest the following changes:

a) to add some changes in the Introduction: Generally I suggest to change introduction and write more about adolescents lifestyle and health. I suggest to add a lifestyle definition and its components in adolescents.  I see a lack of balance between descriptions of socioeconomic position and health and healthy lifestyle. I suggest to begin with some general information concerning adolescents health determinants and after that explain why a socioeconomic position is so important. I suggest to use one of the ecological theory of health or development as a theoretical becground. It will be easier to understand that the socioeconomic posiotion could be one of the direct or indirect aspects which influence on adolescents health. 

-       We have improved the introduction section. As suggested, we present an ecological model of the social determinants on health and better explain how socioeconomic position affects health though material, psychological and behavioral mechanisms. Therefore, the direct and indirect effects of socioeconomic position are placed in a theoretical framework. In addition, we have included a definition of health behaviors and lifestyles, and describe in detail how these dimensions can explain socioeconomic inequalities in health. We appreciate this suggestion and believe that the revised version of the manuscript presents a more complete description of the theoretical background.

1. line 85 - to add a separate paragraf with the aims (subtitle)

-       A space and subtitle for “objectives” have been introduced. In addition, we have modified the aims and hypothesis, presenting them accordingly with the model proposed in the introduction of the revised manuscript.

b) to add some changes in the Matherials and Methods
1. line 123 - Authors should explain why it is better to use GHS than separate dimensions.

-       We thank you for the comment. In response to it we have completed the rationale for employing a global health score in the suggested place.

2. line 130 to add Global score of healthy lifestyle after Thus,

-       Again thanks for the comment. We agree with the reviewer that this sentence was incomplete. Therefore, this change has been done.

c)  in the Discusion - I suggest to write more about the limitations of this paper. I suggest a separate paragraf on the end of the paper.

-       As suggested, we have enriched the limitations of the study, reflecting about the need to analyze the invariance across time and space of the global scores employed in this research, and also the invariance of the proposed models across different populations (thus, testing the examined models across sex, age, ethnicity, and socioeconomic status). Moreover, we have also included in limitations the need of further exploring other factors that might have an effect on health inequalities at different ecological levels.

We hope that all the changes described have contributed to address the reviewers’ comments, in the sense that implications and relevance of the study can now more easily be judged thanks to the explicit statement of research questions and the improvements made specially in the introduction section.

Round 2

Reviewer 1 Report

I am pleased with the authors' response to previous concerns. This will make a nice addition to the literature.